# Earley-Driven Dynamic Pruning for Efficient Structured Decoding

Xintong Sun [* 1]   Chi Wei [* 2]   Minghao Tian [1]   Shiwen Ni [2]

## Abstract

Large Language Models (LLMs) have shown remarkable capabilities, yet ensuring their outputs conform to strict structural or grammatical constraints remains challenging, which is critical in function calls and domain-specific language (DSL) generation. Constrained decoding with context-free grammar is a flexible approach to guarantee LLMs' adherence to a specific format by dynamically building a token logits mask. However, creating this mask requires checking the validity of all tokens in the LLM vocabulary at every decoding step, which often incurs significant overheads in existing constrained decoding engines. To address this challenge, we propose ZapFormat, a novel dynamic pruning strategy based on the Earley algorithm that identifies and eliminates invalid or redundant Earley states in real-time, significantly reducing memory occupation of the Earley algorithm's states. This further enables us to use a state cache to speed up structured generations on a large number of queries. We implemented ZapFormat in a new constrained decoding engine called Formatron which also incorporates existing optimizations. Through comprehensive experiments on structured generation tasks, including JSON generation, JSON Schema validation, and semantic parsing, we demonstrate that Formatron not only consistently maintains high-precision compliant outputs but also achieves significant improvements in **inference speed up to 2x compared to state-of-the-art implementations**. More importantly, Formatron is generally applicable across various LLM architectures. We release Formatron as open source at https://github.com/Dan-wanna-M/formatron.

## 1. Introduction

In recent years, Large Language Models (LLMs) have demonstrated remarkable progress, achieving breakthrough advances across multiple frontier domains including natural language processing (OpenAI & Sandhini Agarwal, 2024), code generation(Chen et al., 2021; Wang et al., 2021), and robotic control (Liu et al., 2023; OpenAI, 2024). As their applications continue to expand, a critical research challenge has emerged: how to guide models to generate text that precisely adheres to required grammatical and formatting specifications. In numerous practical applications, the structural conformity of output text is paramount (Beurer-Kellner et al., 2023; Lundberg et al., 2023). For instance, in function calling or external tool interactions, systems typically require generated text to comply with specific parseable formats (e.g., JSON) , imposing heightened demands on LLMs (Shin et al., 2021; Roy et al., 2024; Fang et al., 2023).

Constrained decoding (Deutsch et al., 2019; Kuchnik et al., 2023) is one of the prevalent technical approaches to tackle this challenge, which filters out tokens that violate grammatical requirements by screening the entire vocabulary at each decoding step, thereby ensuring model outputs conform to specified formal grammars. Among various constraint forms, context-free grammar (CFG) (Chomsky, 1956) is regarded as a flexible and universal constraint format due to its robust descriptive capabilities across multiple languages and structures. However, when applying these methods to large language models, several common challenges emerge:

1. **Computational Overhead from Large Vocabularies and Complex Grammars**. To ensure each newly decoded token correctly follows CFG rules, it is typically necessary to continuously assess the compatibility of all candidate tokens in the vocabulary and maintain the grammar parsing stack or state in real-time. For scenarios involving large vocabularies, long sequences, or complex grammars, this can lead to increases in computational and memory consumption (Geng et al., 2023; Beurer-Kellner et al., 2024).

2. **State Redundancy and Maintenance Complexity**. Decoding requires storing and updating multiple parsing states, including information that may branch or backtrack. When state quantities rapidly accumulate, the timely elimination of repetitive, invalid, or unused states becomes crucial (Willard & Louf, 2023). If not handled properly, these redundant

---

[*]Equal contribution [1]Department of Computer Science, Rice University, Texas, the United States [2]Shenzhen Institutes of Advanced Technology, Chinese Academy of Sciences, Shenzhen, China. Correspondence to: Shiwen Ni <sw.ni@siat.ac.cn>.

*Proceedings of the 42nd International Conference on Machine Learning*, Vancouver, Canada. PMLR 267, 2025. Copyright 2025 by the author(s).

states can trigger cache misses, increase memory usage, and ultimately result in decreased inference speeds (Opedal et al., 2023).

We propose **ZapFormat**, a novel **dynamic pruning** strategy based on the Earley algorithm, which serves as the core component of our new constrained decoding engine **Formatron**. This approach identifies and eliminates invalid or redundant Earley states in real-time, significantly reducing memory occupation. Formatron integrates ZapFormat with state-of-the-art optimizations to achieve both grammatical compliance and computational efficiency.

The key insight behind our approach is that during the parsing process, many intermediate parsing states become "obsolete" or "dead" – meaning they no longer contribute to finding valid parsing paths but continue to consume memory resources. Traditional parsing algorithms, including the widely-used Earley algorithm, tend to accumulate these unnecessary states throughout the decoding process, leading to memory bloat and reduced performance.

Our core mechanism continuously tracks the utility of parsing states during decoding, allowing **Formatron** to swiftly identify and discard these dead states while maintaining only the states useful for parsing. For instance, consider parsing the input "aa" with the grammar $(A \rightarrow AB \mid B; \ B \rightarrow a)$. In the baseline Earley parser, Earley Set 1 (after scanning the first 'a') retains four states: the completed $(B \rightarrow a\bullet; 0)$, active items $(A \rightarrow \bullet AB; 0)$ and $(A \rightarrow B\bullet; 0)$, and the predicted $(B \rightarrow \bullet a; 1)$. However, the completed state $(B \rightarrow a\bullet; 0)$ becomes obsolete once subsequent parsing states (e.g., Earley Set 2's $(A \rightarrow AB\bullet; 0)$) no longer reference its start position. **Formatron**'s dynamic pruning removes such dead states immediately, reducing memory footprint and lowering overall state counts. This approach ensures that only relevant states propagate, streamlining the parsing process without compromising grammatical adherence.

Experimental evaluations on structured generation tasks demonstrate Formatron's efficacy. Experimental results demonstrate that while meeting strict format output requirements, this method effectively reduces memory usage and accelerates inference, providing an efficient and universal technical pathway for structured generation in large language models. Our contributions are as follows:

- We propose a dynamic pruning method based on the Earley algorithm, which significantly reduces memory usage during inference and improves computational efficiency through real-time cleanup of invalid states.

- We design and implement a new constrained decoding engine, Formatron, integrating dynamic pruning with existing optimization techniques. Across multiple structured generation tasks (such as JSON, JSON Schema, and semantic parsing), it maintains high-precision output while achieving up to 2x inference speed improvements.

- We demonstrate the universality of the Formatron engine, showcasing its broad applicability across various language model architectures, efficiently supporting structured generation tasks in both large-scale language models and task-specific applications.

## 2. Related Work

Constrained Decoding is a method that ensures text generated by Large Language Models (LLMs) conforms to specific formats or task requirements. Within the architecture of LLMs, tokens serve as atomic processing units that mediate between raw text input and generated output. While each token represents a fixed-length character sequence, this representation often fails to preserve linguistic integrity—either by splitting semantically coherent units, syntactically meaningful structures, or fragmenting multi-byte Unicode characters (Wang et al., 2020). For instance, as discussed in (Beurer-Kellner et al., 2024), a standard JSON string like "I am Van" might be tokenized into <" I am > and < Van" >, which disrupts both semantic and syntactic coherence. These tokenization limitations pose significant challenges for structured text generation tasks, especially in applications requiring strict syntactic compliance or fine-grained semantic control, such as output requirements for a valid JSON format. Prior work has addressed similar challenges in semantic parsing (Xiao et al., 2016) and syntactic code generation (Yin & Neubig, 2017), demonstrating the importance of incorporating structural constraints during text generation.

Several works (Beurer-Kellner et al., 2023; Lundberg et al., 2023; Willard & Louf, 2023) use regular expressions for constrained decoding, but the approach's expressiveness is also limited to regular expressions, which excludes all formats that can only be described by CFG. These formats include json grammar, json schema, almost all programming languages, and more. To support full CFG, several studies (Scholak et al., 2021; Poesia et al.; Geng et al.) utilise a parser and scanner running in parallel with the LLM, and then calculate online which tokens are valid continuations at each step. However, these approaches incur a relatively high inference overhead, and in the worst case, they must examine a nontrivial subset of vocabulary at each step.

Recently researchers have worked on achieving highly effective and efficient constrained decoding. Beurer-Kellner et al. (2023) use precomputation, speculative decoding and opportunistic masks to achieve minimally invasive and efficient constrained decoding. Koo et al. proposed an approach based on automata theory to provide an efficient closed

form solution for regular languages. Recent work XGrammar (Dong et al., 2024) significantly accelerates constrained decoding by classifying the tokens in the vocabulary into context-independent and context-dependent tokens, which enables effective usage of precomputable adaptive context-independent token masks. To further accelerate constrained decoding, our work investigates a dynamic pruning strategy for the Earley algorithm that is able to identify and eliminate invalid or redundant Earley states in real-time, thus significantly reducing the memory footprint of the Earley algorithm states. This further enables us to leverage Earley state caching to accelerate structured generation.

## 3. Preliminaries

### 3.1. Context-Free Grammar

In computer science, Context-Free Grammar (CFG) represents a crucial grammar type commonly used to describe programming language syntax. *To understand CFGs intuitively, think of them as a set of rules that define how to construct valid sentences or structures, similar to how grammatical rules in natural language define valid sentence constructions.*

A CFG consists of a set of rules (or productions), where each rule has a non-terminal symbol on the left side and a sequence of terminal symbols and non-terminal symbols on the right side. *Here, non-terminal symbols represent abstract structural components (like "noun phrase" in linguistics), while terminal symbols are the actual characters or words that appear in the final text.* Each rule takes the form:

$$A \rightarrow \alpha$$

Where $A$ is a non-terminal symbol and $\alpha$ is a sequence composed of terminal and non-terminal symbols. For example, a simple addition grammar might have the following rule:

$$Expression \rightarrow Expression + Term \quad | \quad Term$$

This grammar rule indicates that an expression can be composed of either another expression plus a term, or simply a term.

In practical applications, CFGs serve as the backbone for parser design in compiler construction and natural language processing. Their formal nature provides a clear framework for analyzing and processing structured input, whether in programming languages or natural language text. For constrained text generation with large language models, CFGs act as blueprints that specify exactly what constitutes valid output, for instance, to ensure that generated JSON follows proper syntax or that code snippets comply with programming language rules.

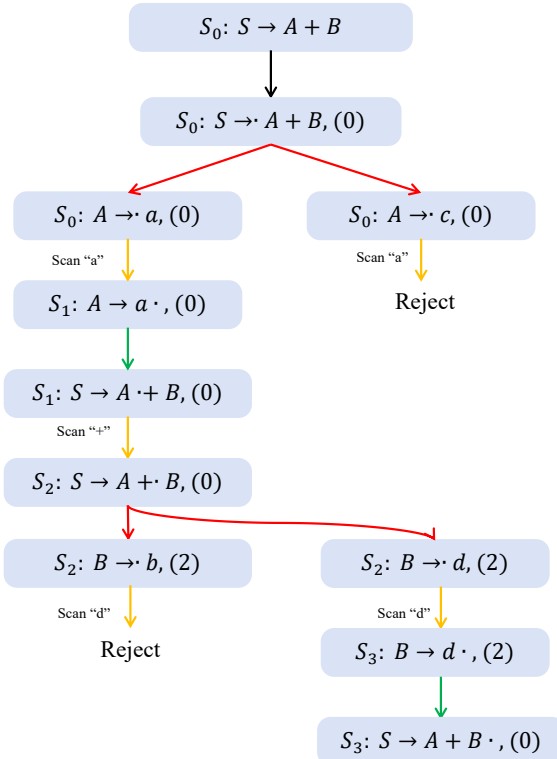

*Figure 1.* Early parse. This diagram provides a detailed illustration of the Earley parsing process for the input string 'a + d' based on the grammar $S \rightarrow A + B, A \rightarrow a|c, B \rightarrow b|d$. In the diagram, red arrows indicate Predict operations; yellow arrows represent Scan operations; and green arrows denote Complete operations.

### 3.2. Earley's Algorithm

The Earley algorithm represents a dynamic programming approach to parsing context-free grammars, notable for its linear time and space complexity for all LR(k) grammars (Leo, 1991), and its polynomial-time $O(n^3)$ parsing capability for ambiguous grammars. Unlike simpler parsing methods that might get stuck or fail when encountering complex grammar structures, the Earley algorithm can handle any context-free grammar, making it particularly valuable for real-world applications where grammar complexity varies significantly. It not only avoids the exponential complexity that plagues many other parsing approaches but also maintains the capability to parse all context-free grammars.

The algorithm maintains a sequence of state sets $S[0], S[1], \ldots, S[n]$, where each set $S[i]$ contains Earley items representing all valid partial parses at position $i$ in the input string. Think of these state sets as snapshots of all possible ways the parser could interpret the input up to each position—similar to how a human reader might consider multiple interpretations of a sentence while reading it word by word. Each Earley item within these sets takes the form

$(X \rightarrow \alpha \bullet \beta, j)$, where $X \rightarrow \alpha\beta$ is a grammar rule with $\alpha$ and $\beta$ being sequences of terminals and non-terminals, the dot ($\bullet$) indicates the current parsing position within the rule and $j$ indicates where in the input string this rule started being applied. The dot can be understood as a bookmark showing "we've successfully matched everything before this point, and we're looking for what comes after."

The algorithm parses the input string through three fundamental operations:

The **prediction operation** handles states of the form $(X \rightarrow \alpha \bullet Y\beta, j)$ where $Y$ is a non-terminal symbol; it adds all rules starting with $Y$ to the current state set $S(k)$, where $k$ is the current position.

The **scanning operation** $(X \rightarrow \alpha \bullet a\beta, j)$ where $a$ is a terminal symbol; if $a$ matches the current input symbol, it adds the state $(X \rightarrow \alpha a \bullet \beta, j)$ to the next state set $S(k+1)$.

The **completion operation** activates when encountering states where the dot has reached the end $(X \rightarrow \gamma \bullet, j)$, where $\gamma$ represents the portion of the rule that has been fully matched up to this point; it finds all states in $S(j)$ where the dot precedes $X$ (like $Y \rightarrow \alpha \bullet X\beta, i$), and adds the advanced state $(Y \rightarrow \alpha X \bullet \beta, i)$ to $S(k)$. Importantly, each state set maintains only unique states without duplicates.

Consider a simple grammar: $S \rightarrow AB$, $A \rightarrow a$, $B \rightarrow b$ processing the input "ab" with the start rule $S \rightarrow \bullet AB$. Through prediction, it adds $A \rightarrow \bullet a$. Upon scanning 'a', it creates $A \rightarrow a\bullet$ in $S[1]$, leading to $S \rightarrow A \bullet B$. The process continues until reaching $S \rightarrow AB\bullet$ in the final state set, confirming a successful parse. This simple example demonstrates how the algorithm systematically explores parsing possibilities: it predicts what could come next, scans to match actual input, and completes patterns when they're fully recognized.

### 3.3. LLM Constrained Decoding

Large Language Models (LLMs) like GPT-4, Llama, and Mistral generate text in an auto-regressive manner: at each step, the model predicts the next token based on previously generated tokens (or input prompt). Specifically, the model outputs a *logits* vector over its vocabulary, which is converted into a probability distribution through the `softmax` function for token sampling.

When generating text that must conform to specific syntactic structures or formats (e.g., JSON, SQL queries, or templated text), directly sampling from the model's probability distribution alone may not guarantee valid outputs. Constrained decoding addresses this challenge by applying a *logits mask* before token sampling. This process sets the logits of invalid tokens that violate output formats to $-\infty$, effectively zeroing their probabilities after `softmax`, ensuring only

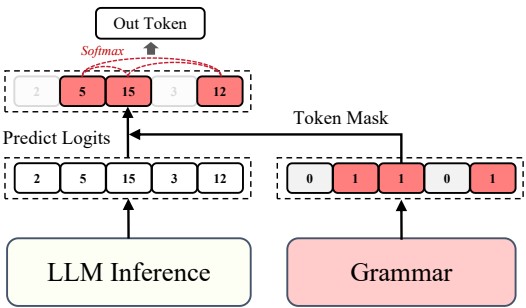

*Figure 2.* Constrained decoding. Constrained decoding can be achieved by masking the illegal tokens at the current step.

valid tokens can be sampled.

To illustrate, consider generating a JSON structure with a simple CFG `JSON -> "{" PairList "}"`, where `PairList` represents key-value pairs. During generation, after producing '{', an Earley parser identifies that only '"' (for starting a string) or '}' (for empty objects) are valid next tokens. The constrained decoding step then masks all other tokens' logits to $-\infty$, restricting sampling to only these valid tokens. This ensures the generated text strictly adheres to the specified grammar.

## 4. Formatron

### 4.1. Motivation

Although the Earley algorithm can parse any context-free grammar and possesses theoretical universality, it often encounters significant memory and computational overhead in practical applications. When the input length is $n$, the Earley parser constructs a set $S[i]$ at each position $i$ (from 0 to $n$). For scenarios involving large inputs or numerous grammar rules, this scale often leads to excessive memory consumption.

Furthermore, to perform backtracking and check expandability during the **Complete** steps, the parser typically retains all previous sets. However, in many cases, certain sets or items become obsolete in subsequent steps: for instance, when parsing JSON objects, once a key-value pair has been fully recognized, there is no need to continue tracking its associated states. Similarly, terminals are often defined using regular expressions and implemented as finite state machines (FSM) for matching; once a terminal's match is completed, the corresponding FSM instance has fulfilled its recognition purpose, and retaining it further only unnecessarily occupies memory resources. In modern computer architectures, these "dead" or "idle" states result in numerous ineffective accesses, increasing the probability of cache misses, thereby slowing down program execution. More

formally, we can define the rules that can create such "dead" states in parser as **High-level Regular Rule**(HRR) in CFG: let the rule's LHS symbol be $A$, then the rule is HRR if the rule satisfies one of the following forms:

$$\{A \to c, A \to Ba, \ A \to \epsilon\}$$

where $A, B$ are non-terminal symbols, $B$ is not ambiguous, $a, c$ are terminal symbols and $\epsilon$ denotes the empty string. Intuitively, HRR creates "dead" states because after $c, \epsilon, B$ are parsed, we technically no longer need to record the states before or during $c, \epsilon$ or $B$'s parsing, while existing parsing algorithms retain them by default.

When the Earley algorithm is employed for **constrained decoding** in conjunction with large language models (LLMs), the aforementioned issues become even more severe. Unlike traditional Earley algorithm processing a **static input sequence** "left-to-right", constrained decoding needs to process all tokens in the vocabulary along with already generated tokens so that we can construct a **logits mask** at each decoding step to filter out candidate tokens that violate syntactic constraints. This requires the Earley parser to **real-time** update and return viable tokens after each LLM decoding step; however, if the parser retains numerous irrelevant states or sets and fails to promptly clean up "dead" or "idle" Earley items, it slows down the decoding process.

To address this, we propose a **dynamic pruning** strategy that aims to minimize redundant storage and repeated computations of ineffective sets while preserving the theoretical completeness of the Earley algorithm. In simple terms, this strategy "online" tracks which sets and states may still be referenced in subsequent steps during the parsing process, and promptly discard items that no longer contribute, significantly reducing memory usage and accelerating parsing speed.

## 4.2. ZapFormat

This section introduces ZapFormat, a novel method for tracking dependencies among Earley items (Earley, 1970) and applying a reachability-based pruning strategy in real time. By maintaining a *dependency graph*, we can effectively remove items that will not contribute to any valid parse, thereby reducing the overall number of states.

### 4.2.1. DEPENDENCIES

To enable more effective tracking of parsing progress and dependencies, we extend the traditional Earley item notation to $(A \to \alpha \bullet \beta, i, j)$ in this enhanced representation, $(A \to \alpha \bullet \beta)$ denotes a production rule from the grammar, while the span $[i, j]$ precisely captures $\alpha$'s coverage of the input sequence from position $i$ up to (but not including) $j$. Based on the extended representation, we proceed to define dependencies. Three types of inter-item dependencies natu-

rally arise from the Earley algorithm's operations: **Predict**, **Scan**, and **Complete**.

**Predict Dependency**. When an item $p = (A \to \alpha \bullet B\beta, i, j)$ triggers the prediction of all rules $B \to \gamma$ in the grammar, each new *predict* item $q = (B \to \bullet\gamma, j, j)$ is said to *depend* on $p$. We then say $D_{\text{pred}}(q, p) \iff B \to \gamma \in p$ and $p = (A \to \alpha \bullet B\beta, i, j)$.

**Scan Dependency**. A SCAN operation moves the dot past a terminal symbol that matches the current input token. If the next input token is $a$ and we have an item $p = (A \to \alpha \bullet a\beta, i, j)$, then scanning yields $q = (A \to \alpha a \bullet \beta, i, j+1)$. We then say: $D_{\text{scan}}(q, p) \iff input[j] = a$ and $p = (A \to \alpha \bullet a\beta, i, j)$.

**Complete Dependency**. The completion operation occurs when we have fully parsed a nonterminal symbol in the grammar. Specifically, when an item has its dot at the end, indicating a complete derivation of a nonterminal B, it can be used to advance all items whose postdot nonterminal is its left hand side. Suppose $p = (B \to \gamma \bullet, k, j)$, $q = (A \to \alpha \bullet B\beta, i, k)$, and we form the new item $r = (A \to \alpha B \bullet \beta, i, j)$. We then say $D_{\text{comp}}(r, p, q) \iff p = (B \to \gamma \bullet, k, j)$ and $q = (A \to \alpha \bullet B\beta, i, k)$.

### 4.2.2. DEPENDENCY GRAPH

To efficiently track these dependencies, we construct a directed graph

$$G = (V, E),$$

where each vertex $v \in V$ corresponds to an Earley item, and each directed edge $(x, y) \in E$ indicates that $y$ depends on $x$. This graph is maintained dynamically as new items are created.

The dependency graph construction process operates on Earley item sets, maintaining vertices $V$ and edges $E$ to track parsing dependencies. When the parser begins, $V$ is empty and grows as new items are discovered. For each item creation through PREDICT, SCAN, or COMPLETE operations, corresponding edges are inserted to record dependencies from source items. The graph updates dynamically as new Earley item sets $(S_0, S_1, \ldots)$ are formed, ensuring comprehensive dependency tracking throughout the parsing process.

### 4.2.3. REACHABILITY AND DYNAMIC PRUNING

For effective pruning, we first define the *reachability closure*. Note that PREDICT, SCAN, and COMPLETE operations all starts by checking items in the lastly created Earley set. Intuitively, this means the reachability of an item can be defined as whether the item depends on any item in the last set. More formally, given an item in an Earley set, we determine which items should be retained through the following definitions:

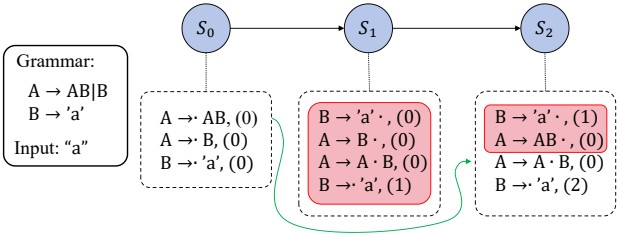

*Figure 3.* Dynamic pruning. Red parts indicate prunable nodes, and green dashed lines represent Complete dependency paths. First,already completed Earley items only lead to modifications on their residing Earley set. Once their residing Earley set is fully processed, we can remove them. In addition, after one Earley set is fully processed, if an Earley set is not referenced by the union of reference chains of all items in the last Earley set, then it can be removed.

**Reachability Closure**: For each item $x \in V$, if there exists a path from $x$ to an item $a$ in the last earley set, we consider this item "reachable". Formally, we define the reachability closure as:

$$\text{Reach}(a) = \{x \in V \mid x \rightarrow^* a\},$$

where $\rightarrow^*$ denotes a directed path (potentially multi-step) from $x$ to $a$.

Active Item Set: We define the active item set $R$ as the union of all reachable items:

$$R = \bigcup_{a \in V} \text{Reach}(a).$$

Only items within this active set need to be retained.

We enhance the original Earley parsing algorithm by introducing a compact operation after the complete phase and before prediction phase, evaluating each item's retention necessity. This phase ordering eliminates useless computations on newly predicted items(which always only depend the items before prediction in the last set). Specifically, the Compact operation examines all items in the current set, eliminating those that do not belong to the active item set $R$.

To ensure both correctness and efficiency of the pruning operations, we implement an incremental update strategy. The active item set is updated after each parsing phase, particularly following the completion and compaction phases. This dynamic maintenance strategy ensures that pruning operations remain responsive to parsing state changes, avoid researching the whole $V$ repeatedly, and maintain algorithmic correctness. By combining forward reachability analysis with dynamic pruning, our algorithm significantly reduces the number of items requiring processing while preserving parsing correctness. This forward reachability-based

dynamic pruning approach not only provides theoretical completeness guarantees but also demonstrates superior performance optimization in practice.

### 4.3. Context-independent Tokens Mask Cache

Inspired by XGrammar's (Dong et al., 2024) context-independent token mask cache, we use a token mask cache method to enhance decoding efficiency in our constrained decoding engine. The core of token mask cache mechanism categorizes tokens into two types: Context-Independent tokens, whose validity can be determined by examining postdot terminals of items in the last Earley set, and Context-Dependent tokens, which require full parsing context. The token mask cache accelerates the parsing process by pre-computing valid and invalid context-independent tokens for each terminal, which are then efficiently stored as bitsets. Note that unlike XGrammar, we do not consider the possible suffix strings of terminals when determining invalid context-independent tokens to save precomputation time. At runtime, the valid and invalid context-independent tokens are retrieved directly from the cache, eliminating redundant computations and thus reducing the overall decoding time.

In scenarios where multiple postdot terminals exist, the token masks from each terminal are merged into a single final token mask. By reducing the number of computations required for token validation through leveraging precompution, the adaptive token mask cache significantly speeds up the decoding process.

### 4.4. Rejection Prefix Optimization

We introduce a prefix-based early rejection mechanism that enhances parsing efficiency by identifying **grammatically impossible input paths** at the earliest stage of the parsing process. The optimization maintains a set of "rejected prefixes" – minimal byte sequences that **definitely preclude valid parses regardless of subsequent input extensions**. These prefixes represent fundamental grammatical violations that cannot be completion. The optimization operates by maintaining and continuously updating these sequences that, when encountered during parsing, immediately indicate the impossibility of a valid parse.

For instance, in a context-free grammar (CFG), if the prefix `"aaac"` constitutes a rejected prefix according to the grammar rules. Once this sequence is detected (e.g., in input `"aaacdefrf"`), no grammatical derivation can produce valid parse trees for any extension of this prefix, regardless of subsequent characters.

When a rejected prefix is detected, the parser can safely discard all extensions of that prefix without state exploration. This set is updated whenever a token is rejected at each decoding step.

## 4.5. Grammar Transformation

To optimize constrained decoding efficiency, we use a grammar transformation framework (Hopcroft & Ullman, 1979a;b). The primary transformation step involves structural optimization, where we identify and eliminate useless rules that cannot contribute to valid derivations, thereby reducing the grammar size without affecting its expressiveness. This is particularly helpful for grammar generated from a high-level format like json schema, where the generator, potentially built by third parties, may fail to remove all unreferenced rules.

Another crucial optimization addresses null rules. We systematically handle rules that can derive empty strings through a three-phase approach: first identifying all null symbols, then generating alternative productions, and finally selectively retaining specific null productions where necessary. This transformation substantially reduces the branching factor during parsing.

## 5. Results

### 5.1. Experimental Setup

All experiments were conducted on a system equipped with an NVIDIA GeForce RTX 3090 (24GB VRAM) and an AMD EPYC 7452 32-core processor. The software environment consisted of PyTorch 2.4.0 and CUDA 12.4, with model inference performed using Transformers v4.48.0. Four pre-trained large language models were employed in this study: google/gemma-2-9b-it (Gemma Team & Shreya Pathak, 2024), meta-llama/Llama-3-8B-Instruct (Dubey et al., 2024), mistralai/Mistral-7B-Instruct-v0.3 (Jiang et al., 2023), and qwen/Qwen2.5-7B-Instruct (Yang et al., 2024), all utilizing half-precision (FP16) inference. For more details of Python libraries, see the appendix A.

**Baselines.** lm-format-enforcer (v0.10.9) (Jiang et al., 2024) implements incremental validation based on syntax tree pre-computation, suitable for structured constraints but with significant memory overhead. outlines (v0.1.13) (Willard & Louf, 2023) employs finite state machines for dynamic masking of invalid tokens, excelling in regular constraints but not directly applicable to context-free grammars. One significant consequence of this limitation is that they can only support a small subset of json schemas (dottxt-ai/outlines contributors, 2023). XGrammar (v0.1.10) (Dong et al., 2024) supports Context-Free Grammars (CFG) through simulating pushdown automata with tree-structured stacks, offering high versatility but introducing parsing latency. It also cannot directly handle left-recursive CFGs (mlc-ai/xgrammar contributors, 2024).

**Test Task.** Geoquery (Davis & Meltzer, 2007) transformation converts natural language queries into FunQL, adhering to fixed predicates and finite entity constraints. JSON Schema (Pezoa et al., 2016) generation produces JSON instances compliant with type, enumeration, and regular expression constraints. JSON Grammar generation creates syntactically valid and semantically coherent nested JSON structures with cross-field dependencies. Data examples are shown in Appendix D.

**Evaluation Metrics.** Throughput represents a fundamental metric in the assessment of LLM constrained decoding performance, defined as the ratio of constraint-satisfying tokens generated to temporal duration, expressed in tokens per second (Token/s). This measurement provides insights into computational efficiency and resource utilization efficacy.

### 5.2. Experimental Results

We conducted a comprehensive evaluation of the formatron engine across four mainstream large language models. Table 1 presents a performance comparison of different approaches under various constraint types:

**Performance Advantage Analysis.** The experimental results robustly validate the efficacy of formatron. In most tested scenarios, formatron achieved significant performance improvements compared to baseline methods. These enhancements can be attributed to formatron's core innovation: dynamically pruning invalid or redundant Earley states, substantially reducing memory consumption during constraint parsing. Coupled with an efficient caching mechanism, formatron consistently maintains high-throughput output. These empirical data comprehensively demonstrate formatron's performance superiority in constrained decoding tasks.

**Robustness Analysis.** In terms of model robustness, formatron exhibits exceptional cross-architectural adaptability. By testing across models with different scales and architectures, formatron consistently maintains stable performance, verifying its characteristic of being generally applicable across various LLM architectures. From 9B to 7B model scales, formatron demonstrates consistent high-efficiency performance.

Regarding task robustness, formatron excels in diverse constrained decoding tasks. From JSON generation to JSON Schema validation and semantic parsing, formatron consistently produces high-precision compliant outputs. This cross-task stability comprehensively substantiates the reliability and generalizability of the formatron technical approach.

**Comparative Method Analysis.** Formatron possesses advantages compared to existing methods. While outlines only supports regular grammars through finite state automata, formatron manages to support full context-free grammar with Earley algorithm and achieves superior performance

*Table 1.* Comparative Throughput Performance of Different Methods in Constraint Parsing Tasks (tokens/s). Here, '-' indicates that the method is not applicable to the task, and bold indicates the best result. All experimental results were obtained in the same local environment. lm-format represents lm-format-enforcer. Json_s represents Json_Scheam and Json_g represents Json_Grammar.

| Model | Method | geoquery | json_s | json_g |
|---|---|---|---|---|
| Gemma | lm-format | - | 60.08 | 22.95 |
| | Outlines | - | 61.32 | - |
| | Xgrammar | 616.66 | 1473.99 | **10245.04** |
| | Formatron | **12174.68** | **7943.34** | 8668.69 |
| Llama3 | lm-format | - | 120.56 | 47.79 |
| | Outlines | - | 114.10 | - |
| | Xgrammar | 2758.98 | 2796.36 | 7757.15 |
| | Formatron | **6700.87** | **8207.85** | **8535.64** |
| Mistral | lm-format | - | 576.11 | 372.55 |
| | Outlines | - | 598.98 | - |
| | Xgrammar | 5926.64 | 8421.43 | **15273.72** |
| | Formatron | **11703.00** | 10639.87 | 12046.54 |
| Qwen | lm-format | - | 45.31 | 21.26 |
| | Outlines | - | 112.10 | - |
| | Xgrammar | 1725.86 | 2037.26 | 7290.30 |
| | Formatron | **6399.68** | **9234.08** | **9811.02** |

*Table 2.* Throughput Comparison of Different Models and Methods Across Multiple Runs. lm-format represents lm-format-enforcer.

| Model | Method | Number of Runs | | | |
|---|---|---|---|---|---|
| | | 1 run | 3 run | 5 run | 10 run |
| Gemma | lm-format | 60.08 | 63.48 | 66.15 | 75.84 |
| | Outlines | 61.62 | 61.66 | 62.00 | 62.13 |
| | Xgrammar | 1473.99 | 1577.05 | 1564.97 | 1551.51 |
| | Formatron | **7943.34** | **10609.08** | **11323.63** | **11993.89** |
| Llama3 | lm-format | 120.56 | 121.51 | 139.99 | 159.34 |
| | Outlines | 114.10 | 104.71 | 105.13 | 103.44 |
| | Xgrammar | 2796.36 | 2533.45 | 2578.91 | 2629.54 |
| | Formatron | **8207.85** | **10271.73** | **11018.91** | **11945.94** |
| Mistral | lm-format | 576.11 | 723.72 | 794.30 | 796.68 |
| | Outlines | 598.98 | 634.63 | 649.78 | 641.22 |
| | Xgrammar | 8421.43 | 7673.90 | 7699.77 | 7885.20 |
| | Formatron | **10639.87** | **13522.77** | **13841.85** | **14424.69** |
| Qwen | lm-format | 45.31 | 41.89 | 46.65 | 49.47 |
| | Outlines | 112.10 | 89.82 | 90.13 | 90.22 |
| | Xgrammar | 2037.26 | 2247.04 | 2299.04 | 2284.37 |
| | Formatron | **9234.08** | **10537.26** | **11397.31** | **12202.48** |

over outlines on regular grammars. Unlike lmformatenforcer's naive input token prefix cache, formatron's state cache with dynamic pruning greatly increases cache hit rate and decreases memory occupation, since multiple different input token sequences can correspond to the same pruned Earley states. Notably, XGrammar, developed concurrently with our work (published within three months of our paper submission), represents a contemporary approach in this domain. By combining our novel algorithm with optimizations proposed by XGrammar, formatron achieves competitive performance across most tasks, with XGrammar showing superior performance in certain specific scenarios (e.g., geoquery on Gemma and Json_s on Mistral), while formatron excels in the majority of other tasks without incurring more precomputation costs. These advantages substantially enhance formatron's processing efficiency in constrained decoding tasks.

To further validate the effectiveness and reliability of our proposed method, we conducted two additional experiments addressing potential concerns about our evaluation methodology. First, we performed a comprehensive analysis of whole pipeline component performance. Second, we evaluated output quality using accuracy metrics to ensure that our efficiency gains do not come at the expense of generation quality. The detailed results of these supplementary experiments are presented in Appendix B.

### 5.3. Multiple Runs

To mitigate the impact of random factors, we conducted multiple runs of experiments on **Json_Schema**. To conduct multiple runs, we utilized LLM for data augmentation. For further details, please refer to the appendix E. The experimental results (shown in Table 2) demonstrate that our proposed Formatron method significantly outperforms existing baseline approaches in terms of throughput. Through systematic evaluation across different numbers of runs, we observed several key trends:

First, Formatron exhibits remarkable performance advantages in single-run scenarios. Across all tested models, Formatron achieves substantially higher throughput compared to baseline methods, with particularly notable advantages in the Mistral and Qwen models. This performance enhancement can be primarily attributed to Formatron's dynamic pruning mechanism, which effectively reduces redundant operations in computation paths.

Furthermore, in multi-run scenarios, we observed an interesting performance evolution pattern. Formatron demonstrates progressive throughput improvement as the number of runs increases, showing consistent performance gains from three runs to five and ten runs. In contrast, baseline methods show limited improvement in multi-run scenarios, exhibiting stability but lacking breakthrough performance. This phenomenon demonstrates Formatron's performance stability and robustness across extended operations.

*Table 3.* Experimental results of ablation. Here, "-& cache" indicates the ablation of the cache in addition to the ablation of pruning.

| LLM | Method | Number of runs | | |
|---|---|---|---|---|
| | | 3 run | 5 run | 10 run |
| Gemma | Formatron | 10609.08 | 11323.63 | 11993.89 |
| | -pruning | 6981.35 | 6628.69 | 7674.78 |
| | - & cache | 3595.48 | 3038.70 | 3365.96 |
| Llama3 | Formatron | 10271.73 | 11018.91 | 11945.94 |
| | -pruning | 6269.24 | 7064.69 | 7619.57 |
| | -& cache | 3464.36 | 3438.40 | 3445.73 |
| mistral | Formatron | 13522.77 | 13841.85 | 14424.69 |
| | -pruning | 9727.00 | 9824.90 | 9707.45 |
| | -& cache | 8845.90 | 8204.43 | 7669.42 |
| qwen | Formatron | 10537.26 | 11397.31 | 12202.48 |
| | -pruning | 8635.20 | 9476.70 | 10048.19 |
| | -& cache | 4189.57 | 4180.18 | 4196.09 |

*Table 4.* Maximum memory usage comparison during constrained decoding.

| Model | Method | Max Memory Usage (MB) |
|---|---|---|
| Llama3 | Formatron | 1635.92 |
| | w/o pruning | 1655.48 |
| Mistral | Formatron | 1519.09 |
| | w/o pruning | 1530.77 |

## 5.4. Ablation Study

To investigate the contribution of each component in Formatron, we conducted ablation experiments by removing the pruning and caching mechanisms. Table 5.4 presents the results across different models and run configurations.

The ablation study reveals that removing the pruning mechanism (-pruning) results in significant performance degradation, with throughput reductions of 30-50% for most models, demonstrating its crucial role in optimization. The further removal of caching (-& cache) generally leads to additional performance deterioration.

In addition, even with both pruning and caching ablated, our method still outperforms several baselines, demonstrating the substantial contributions of other optimizations discussed earlier, including prefix rejection and grammar transformation.

On the other hand, in order to validate our claims regarding memory reduction, we conducted additional experiments measuring the maximum memory usage (in MB) during constrained decoding (Table 4). The results show that our pruning mechanism reduces memory consumption. These results confirm that our approach successfully reduces memory footprint by eliminating redundant Earley states.

## 6. Conclusion

We present Formatron, an efficient LLM constrained decoding engine that achieves 1.5-2× speedup over state-of-the-art method. Our Formatron contains two core innovations: (1) The proposed ZapFormat algorithm based on the identifications of HRR in CFG, which reduces memory usage via dynamic Earley state pruning based on dependency reachability analysis; (2) A hybrid optimization framework combining grammar-aware token masking with prefix rejection. Evaluations across JSON generation and semantic parsing tasks demonstrate consistent performance improvements while maintaining 100% structural compliance, establishing new efficiency benchmarks for grammar-guided LLM decoding.

## Acknowledgements

Shiwen Ni was supported by GuangDong Basic and Applied Basic Research Foundation (2023A1515110718 and 2024A1515012003), China Postdoctoral Science Foundation (2024M753398), Postdoctoral Fellowship Program of CPSF (GZC20232873) and Shenzhen Major Science and Technology Project (KCXFZ20240903094007010).

## Impact Statement

This study introduces methodological improvements for constrained decoding in large language models, with particular emphasis on computational efficiency gains for structured text generation tasks. Our approach enhances the reliable generation of grammatically valid outputs in practical applications ranging from automated API call generation to standardized data formatting. The proposed techniques raise ethical considerations common to language model technologies, particularly regarding their potential deployment in automated decision-making systems. While the improved efficiency may expand access to structured generation tools across various sectors, appropriate safeguards are still necessary to mitigate risks associated with uncontrolled automation. It is important to note that the main technical advance—refining grammatical constraint enforcement mechanisms—does not introduce novel ethical concerns beyond those already present in conventional language model applications. Researchers and practitioners adopting these methods should maintain established responsible innovation practices, including rigorous output verification and maintenance of human oversight protocols.

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

# A. Appendix: Python Library Versions

This appendix provides a comprehensive list of Python libraries and their respective versions used in our project. Documenting the exact versions of these dependencies is crucial for ensuring the reproducibility of our work. By maintaining a consistent environment with the specified versions, readers can accurately replicate our experiments and avoid potential issues arising from version discrepancies. Below, we present the libraries and their versions in a tabular format for easy reference.

*Table 5.* Python Libraries and Versions

| Package | Version | Package | Version | Package | Version |
|---|---|---|---|---|---|
| accelerate | 1.2.1 | aiohappyeyeballs | 2.4.4 | aiohttp | 3.11.11 |
| aiosignal | 1.3.2 | airportsdata | 20241001 | annotated-types | 0.7.0 |
| astunparse | 1.6.3 | attrs | 24.3.0 | cloudpickle | 3.1.1 |
| datasets | 3.2.0 | dill | 0.3.8 | diskcache | 5.6.3 |
| dnspython | 2.6.1 | einops | 0.8.0 | expecttest | 0.2.1 |
| flash-attn | 2.7.3 | frozendict | 2.4.6 | frozenlist | 1.5.0 |
| fsspec | 2024.6.1 | general_sam | 1.0.1 | huggingface-hub | 0.27.1 |
| hypothesis | 6.108.4 | iniconfig | 2.0.0 | interegular | 0.3.3 |
| jsonpointer | 2.1 | jsonschema | 4.23.0 | jsonschema-specifications | 2024.10.1 |
| lark | 1.2.2 | lintrunner | 0.12.5 | lm-format-enforcer | 0.10.9 |
| mkl-service | 2.4.0 | multidict | 6.1.0 | multiprocess | 0.70.16 |
| nest-asyncio | 1.6.0 | ninja | 1.11.1.1 | numpy | 1.26.4 |
| optree | 0.12.1 | outlines | 0.1.13 | outlines_core | 0.1.26 |
| pandas | 2.2.3 | pluggy | 1.5.0 | propcache | 0.2.1 |
| protobuf | 5.29.3 | pyarrow | 19.0.0 | pybind11 | 2.13.6 |
| pycountry | 24.6.1 | pydantic | 2.10.5 | pydantic_core | 2.27.2 |
| pytest | 8.3.4 | python-dateutil | 2.9.0.post0 | python-etcd | 0.4.5 |
| referencing | 0.36.1 | regex | 2024.11.6 | rpds-py | 0.22.3 |
| safetensors | 0.5.2 | sentencepiece | 0.2.0 | sortedcontainers | 2.4.0 |
| tiktoken | 0.8.0 | tokenizers | 0.21.0 | torch | 2.4.0 |
| torchaudio | 2.4.0 | torchelastic | 0.2.2 | torchvision | 0.19.0 |
| transformers | 4.48.0 | triton | 3.0.0 | types-dataclasses | 0.6.6 |
| typing_extensions | 4.12.2 | tzdata | 2024.2 | xgrammar | 0.1.10 |
| xxhash | 3.5.0 | yarl | 1.18.3 | | |

# B. Additional Experimental Results

To address potential concerns about the completeness of our evaluation and to provide a more comprehensive analysis of our proposed method, we conducted two additional experiments that examine different aspects of performance measurement and output quality assessment.

## B.1. Pipeline Component Analysis: Parsing and Masking Throughput

In response to questions about whether our reported throughput results reflect the performance of the entire generation pipeline, we clarify that the main results presented in our paper focus specifically on the parsing and mask generation stages to provide precise performance analysis of our key technical innovations. However, to address this concern, we conducted additional experiments measuring throughput for the complete pipeline, including throughput without constrained decoding (w/o CD).

Table 6 presents the detailed throughput measurements (JSON objects per second) across different models and methods. While the complete pipeline results show lower throughput due to the inclusion of LLM calls and generation processes, our proposed Formatron method maintains competitive efficiency compared to other constrained decoding approaches, performing nearly as well as the unconstrained baseline (w/o CD).

*Table 6.* Throughput comparison for parsing and masking stages across different models and methods (JSON objects per second)

| Model | Method | JSON/s |
|-------|--------|--------|
| Gemma | w/o CD | 18.16 |
|       | lm-format | 11.75 |
|       | Outlines | 13.56 |
|       | Xgrammar | 17.72 |
|       | Formatron | 18.02 |
| Llama3 | w/o CD | 31.75 |
|       | lm-format | 20.51 |
|       | Outlines | 23.71 |
|       | Xgrammar | 30.42 |
|       | Formatron | 30.42 |
| Qwen | w/o CD | 35.40 |
|       | lm-format | 16.23 |
|       | Outlines | 25.98 |
|       | Xgrammar | 34.34 |
|       | Formatron | 34.46 |

## B.2. Output Quality Assessment: Accuracy Metrics

While a single schema is compatible across multiple constrained decoding libraries, currently no standardized interpretation exists across implementations. JSON Schema, for instance, does not define allowed whitespace patterns or positions within valid JSON structures. Given that most models' tokenizers are sensitive to whitespace character position and type, identical schemas may yield different outputs when processed through different libraries.

To provide a more comprehensive evaluation beyond throughput measurements, we conducted additional experiments to assess output quality using accuracy metrics. This addresses the important question of whether performance gains come at the cost of generation quality.

Table 7 shows the accuracy results for different models and methods across two evaluation scenarios: `json_schema` and `json_grammar`. The results demonstrate that our proposed method achieves competitive performance compared to existing approaches while maintaining the efficiency advantages shown in the main results.

*Table 7.* Accuracy comparison across different models and methods for JSON schema and grammar tasks

| Model | Method | json_schema | json_grammar |
|-------|--------|-------------|--------------|
| Gemma | baseline | 0.73 | - |
|       | lm-format | 0.74 | 0.71 |
|       | Outlines | **0.80** | - |
|       | Xgrammar | 0.76 | 0.71 |
|       | Formatron | 0.73 | **0.74** |
| Llama3 | baseline | 0.47 | - |
|       | lm-format | 0.60 | 0.40 |
|       | Outlines | **0.73** | - |
|       | Xgrammar | 0.69 | 0.47 |
|       | Formatron | 0.67 | **0.48** |
| Mistral | baseline | 0.09 | - |
|       | lm-format | **0.53** | 0.10 |
|       | Outlines | 0.44 | - |
|       | Xgrammar | **0.53** | 0.09 |
|       | Formatron | 0.52 | **0.11** |

These additional experiments confirm that our method maintains competitive accuracy while achieving superior efficiency, thus validating both the effectiveness and reliability of our approach.

## C. Notation and Terminology

To improve the readability of this paper, we provide a comprehensive notation table for reference. The following table contains the key symbols and concepts used throughout our work.

| Symbol | Description |
|---|---|
| $A, B, X, Y$ | Non-terminal symbols in context-free grammar |
| $a, c$ | Terminal symbols in context-free grammar |
| $\alpha, \beta, \gamma$ | Sequences composed of terminal and non-terminal symbols |
| $\varepsilon$ | Empty string |
| $S[i], \ldots, S[n]$ | Sequence of Earley state sets, where $S[i]$ contains items at position $i$ |
| $(X \to \alpha \bullet \beta, j)$ | Earley item notation, where $\bullet$ indicates current parsing position and $j$ indicates starting position |
| $(A \to \alpha \bullet \beta, i, j)$ | Extended Earley item notation, where span $[i, j]$ captures $\beta$'s coverage range |

## D. Task Examples

Below are input-output examples for the three constrained generation tasks:

**Geometry**

```
1  Input: name all the rivers in colorado.
2  Output: answer(river(loc_2(stateid('colorado'))))
```

**JSON Schema and JSON Grammar**

```
1   Input:
2   [
3     {
4       "content": "You are a helpful assistant that answers in JSON. Here's the
          ↪ json schema you must adhere to:\n<schema>\n{'title': '
          ↪ WirelessAccessPoint', 'type': 'object', 'properties': {'ssid': {'
          ↪ title': 'SSID', 'type': 'string'}, 'securityProtocol': {'title': '
          ↪ SecurityProtocol', 'type': 'string'}, 'bandwidth': {'title': '
          ↪ Bandwidth', 'type': 'string'}}, 'required': ['ssid', '
          ↪ securityProtocol', 'bandwidth']}\n</schema>\n",
5       "role": "system"
6     },
7     {
8       "content": "I'm currently configuring a wireless access point for our
          ↪ office network and I need to generate a JSON object that accurately
          ↪  represents its settings. The access point's SSID should be '
          ↪ OfficeNetSecure', it uses WPA2-Enterprise as its security protocol,
          ↪  and it's capable of a bandwidth of up to 1300 Mbps on the 5 GHz
          ↪ band. This JSON object will be used to document our network
          ↪ configurations and to automate the setup process for additional
          ↪ access points in the future. Please provide a JSON object that
          ↪ includes these details.",
9       "role": "user"
10    }
11  ]
12  Output:
13  {
14    "ssid": "OfficeNetSecure",
```

```
15    ” s e c u r i t y P r o t o c o l ” :  ”WPA2−E n t e r p r i s e ” ,
16    ” bandwidth ” :  ”1300 Mbps”
17  }
```

## E. Data Augmentation Workflow

**Text Generation** We use five models (`deepseek-chat`, `gemini-2.0-flash-exp`, `gemini-exp`, `gpt-4`, and `gpt-4o`) to generate text variations. Each model generates 25 variations, resulting in a total of 125 data points. The following prompt is used for text generation:

```
1   Generate  25  v a r i a t i o n s  o f  the  user  message .  Follow  these  g u i d e l i n e s :
2   1 .  MUST  i n c l u d e  a l l  r e q u i r e d  f i e l d s  and  maintain  data  a c c u r a c y .
3   2 .  Vary  s e n t e n c e  s t r u c t u r e s  using  these  t e c h n i q u e s :
4      −  Use  d i f f e r e n t  verb  forms  ( a c t i v e / p a s s i v e ) .
5      −  Apply  paraphrasing  while  keeping  the  same  meaning .
6      −  Change  word  order  when  p o s s i b l e .
7      −  Use  synonyms  for  non−key  terms .
8   3 .  Vary  these  p r o p e r t i e s  c r e a t i v e l y :  { chr ( 1 0 ) . j o i n ( f i e l d _ i n s t r u c t i o n s ) } .
9   4 .  Keep  key  terminology  c o n s i s t e n t  ( names / IDs / e t c ) .
10  5 .  Sound  n a t u r a l  and  c o n v e r s a t i o n a l .
11  6 .  Answer  in  E n g l i s h .
12  7 .  Output  MUST  be  pure  JSON  only  −  no  t e x t ,  comments ,  or  markdown .
13
14  Required  JSON  format :
15  {
16    ” c o n t e n t ” :  [
17      {
18        ” c o n t e n t ” :  ”Example  message  1 . . . ” ,
19        ” r o l e ” :  ” u s e r ”
20      } ,
21      {
22        ” c o n t e n t ” :  ”Example  message  2 . . . ” ,
23        ” r o l e ” :  ” u s e r ”
24      }
25    ]
26  }
27
28  C r i t i c a l  Requirements :
29  −  F i n a l  JSON  MUST NOT  be  t r u n c a t e d .
30  −  Last  a r r a y  item  MUST  end  with  }}]}}  without  a  comma .
31  −  Escape  a l l  double  quotes  i n s i d e  c o n t e n t .
32  −  Ensure  a l l  b r a c k e t s  are  p r o p e r l y  c l o s e d .
33
34  O r i g i n a l  Message  S t r u c t u r e  Reference :  { o r i g i n a l _ m e s s a g e }
```

**Data Processing** The `content` field is extracted from the generated JSON data. Incomplete or malformed data points are removed during initial filtering.

**Quality Evaluation** We use o1 to evaluate the filtered data. The evaluation prompt is as follows:

```
1  You  are  an  e x p e r t  t e x t  q u a l i t y  e v a l u a t o r .
2  For  each  input  t e x t ,  provide  a  JSON  o b j e c t  with  a  ” r e s u l t s ”  a r r a y  c o n t a i n i n g
      ↪  e v a l u a t i o n  o b j e c t s .
3  Each  e v a l u a t i o n  o b j e c t  must  c o n t a i n  e x a c t l y  four  f i e l d s :
4  −  ” textNumber ” :  The  index  of  the  t e x t  ( s t a r t i n g  from  1 ) .
```

```
 5   – "relevance": A relevance score between 0 and 100 (based on schema keywords)
     ↪  .
 6   – "uniqueness": A uniqueness score between 0 and 100 (based on sentence
     ↪  structure).
 7   – "coherence": A coherence score between 0 and 100 (based on semantic flow).
 8
 9   Response format:
10   {
11     "results": [
12       { "textNumber": 1, "relevance": 94, "uniqueness": 85, "coherence": 90 },
13       { "textNumber": 2, "relevance": 95, "uniqueness": 88, "coherence": 92 },
14       { "textNumber": 3, "relevance": 93, "uniqueness": 82, "coherence": 89 }
15     ]
16   }
17
18   Important rules:
19   1. Each text must be evaluated individually.
20   2. The "textNumber" must match the order of the input texts (starting from 1)
     ↪  .
21   3. Scores must be integers between 0 and 100.
22   4. Do not include any additional fields or comments in the JSON response.
23   5. Ensure the response is valid JSON and can be parsed directly.
```

The data is sorted based on evaluation scores (relevance, uniqueness, and coherence). For each schema, the top 100 highest-scoring data points are retained. The final filtered data is saved as a JSON file for downstream use.

