# OpenReview forum: "Earley-Driven Dynamic Pruning for Efficient Structured Decoding"
_ICML.cc/2025/Conference — ICML 2025 poster_

### Official Review · Reviewer_wJ9T · 2025-02-22

**Overall Recommendation:** 3

**Summary:**

LLM can be equipped with a grammar verifier which verifies next token prediction at each step to satisfy grammatical constraints. The key step is to incremental update the grammar state in the parsing algorithm to output the possible next tokens. Given a general form of grammar (CFG), the paper leverages Earley parsing and proposes improvements on the Earley parsing algorithm to achieve efficient next possible token calculations. Compared to available techniques, the paper achieves efficient next possible token calculation efficiency as demonstrated by the high throughput for LLM constrained decoding process.

**Claims And Evidence:**

The paper claims that the following techniques achieve high throughput (high efficiency of the next possible token calculation):
- Overall: yes, higher than existing techniques significantly (Table1)
- By pruning non influencing states: yes, Table 3
- By optimizing cache: yes, Table 3. Although cache optimization is proposed in XGrammar (2024) a detailed comparison to compare the two is missing.
- Rejection Prefix Optimization: No
- Grammar Transformation: No
Although the later two might contribute to the overall throughput improvement.

**Essential References Not Discussed:**

No. However, I think it would be nice to mention that the NLP community has examined neural constrained decoding from 2016 with papers like https://aclanthology.org/P16-1127 and https://arxiv.org/abs/1704.01696

**Experimental Designs Or Analyses:**

- 4.5 helps the performance but is not a strong scientific contribution, from this viewpoint, a baseline is missing where only 4.5 is present in the system to measure throughput.
- It is also important to know the throughput without constrained decoding
- There requires an explanation why 3 run throughput decreases consistently for the proposed techniques.

**Methods And Evaluation Criteria:**

The throughput seems to be a sensible metric for this task.

Two datasets are established benchmarks and fit well to the studied problem. The authors further propose another task that is similar in grammar to json_s, so a sensible benchmark.

**Other Comments Or Suggestions:**

- 4.2.1 and 4.2.2 introduces the format and the associated graph, however, it is not clearly stated how such representations help the pruning, putting which I think would enhance the paper readability.
- As mentioned previously, I think some baselines are missing (or would be better to include): a baseline with only changes from 4.5 and a baseline where no contrained decoding is used.

**Other Strengths And Weaknesses:**

None

**Questions For Authors:**

- How do you explain the decrease about 3 times run?
- Do you have a concrete example in the dataset where you can show techniques in 4.4 helps
- Are there comparisons between the different cache strategies in terms of performance?

**Relation To Broader Scientific Literature:**

- It is related to grammar parsing. The paper is based on Earley parsing algorithm which is a classic parsing algorithm for general CFG grammar.
- It is also related to accelerating LLM throughput which is broadly related to LLM inference speed.

**Theoretical Claims:**

- In Earley algorithm, pruning the particular states as the authors indicate should be able to save memory, eliminate useless states for next token parsing (4.1) thus save time for next possible token calculations.
 - The dependency construction using indices help to identify and prune completed states to help accelerate the next token prediction calculations; although I think it is not clearly stated in the paper how this new presentation helps pruning.
- The cache helps as already shown by XGrammar.
- 4.4 Not sure how much such techniques help in practice: taking the same example as the paper, if "aaac" if an invalid prefix, it would never be generated by constrained decoding as a prefix. The help happens to reject "aaacdefrf" a bit quicker during state calculation.
- 4.5 The claim is right but the benchmarks should more appropriately use 4.5 as baselines.

---

> ### Author Rebuttal · Authors · 2025-03-31
>
> ## Q1: Although cache optimization is proposed in XGrammar (2024) a detailed comparison to compare the two is missing.
>
> We appreciate the reviewer's concern. Section 4.3 addresses this comparison. Both methods categorize tokens into Context-Independent and Context-Dependent types, but we don't use suffix strings to identify invalid context-dependent tokens. For example, XGrammar precomputes all possible suffixes of `{` to reject tokens like `{//ABC` without the parser, while we don't.
> We omit this optimization because:
> 1. There's no theoretical guarantee that it significantly reduces context-dependent tokens across all grammars and vocabularies,
> 2. Precomputation overhead scales with grammar size.
> 3. Formatron already matches or exceeds XGrammar's throughput without it.
>
> That said, this optimization is orthogonal to ours, so it is possible to include the optimization if needed.
>
> ## Q2: Do you have a concrete example in the dataset where you can show techniques in 4.4 helps?
>
> Thank you for your insightful question. When a context-dependent token is rejected by the parser, the bytes from the first to the byte that is rejected by the parser become a rejected prefix. For each unprocessed context-dependent token, if it shares a prefix with rejected prefixes, then it can be immediately rejected. This allows us to reject tokens with common prefixes faster.
>  We collected following examples from experiments:
> - prefix: ` ".`, token: ` "..\..\..\`
> - prefix: `=""`, token: `=""></`
> - prefix: `="<`, token: `="<?=$`
>
> ## Q3: From this viewpoint, a baseline is missing where only 4.5 is present
>
> We agree with your suggestion. Here's the baseline with 4.5 only.
> |Model|Method| json_s|
> |-|-|-|
> |Genma|Formatron|7453.87|
> || only 4.5| 27.90|
> |Llama3| Formatron|7616.25|
> ||only 4.5|50.04|
> |Mistral| Formatron|12828.53|
> ||only 4.5| 264.31|
> |Qwen| Formatron| 8449.92|
> ||only 4.5| 46.69|
>
> ## Q4: How do you explain the decrease about 3 times run?
>
> Thank you for your careful review. **Reviewer YX9o** had the same question. To allow more space for addressing your other concerns in detail, please refer to our response to **Reviewer YX9o: Q3**.
>
> ## Q5: No. However, I think it would be nice to mention that the NLP community has examined neural constrained decoding from 2016
>
> We appreciate the reviewer's suggestion. We will incorporate Xiao et al. (2016) and Yin et al. (2017) in our revised manuscript as important references to this field's early work.
> We did not include them as baselines because:
> 1. Adapting these methods to transformer architecture would require significant engineering modifications and LLM continual pretraining.
> 2. The grammar state maintenance in these approaches would consume excessive memory when applied to LLM.
>
> ## Q6: Sections 4.2.1 and 4.2.2 introduce the format and the associated graph, however, it is not clearly stated how such representations help the pruning. Adding this would enhance the paper's readability.
>
> Thank you for requesting clarification on how the representations facilitate pruning. You're right that sections 4.2.1 and 4.2.2 would benefit from a more explicit connection to the pruning mechanism.
>
> 4.2.1 defines inter-item dependencies and lays the foundation for pruning: for independent items, removing one will not affect Earley actions on others. The enhanced Earley item representation emphasizes that while the same Earley item often spans multiple sets, its dependencies are only affected by its starting and end position. Thus, we need not search all Earley sets in between to find all its dependencies.
>
> 4.2.2 defines the dependency graph directly enables pruning since reachability closure is defined on this graph. A path from `x` to `a` is a dependency chain from `a` to `x`. If `a` (indirectly) depends on `x`, then `x` is reachable from `a`. That's why reachable items must be kept; removing them would disrupt the dependency chains required for correctness.
>
> ## Q7: It is also important to know the throughput without constrained decoding
>
> We appreciate your insightful question. The throughput metric in our manuscript is only applicable when constrained decoding component is present in the pipeline since it only measures the throughput of the constrained decoding component only. To answer your question, we conducted additional experiments on throughput for the entire pipeline, including the throughput without contrained decoding (w/o CD).
>
> |Model|Methods|Json/s|
> |-|-|-|
> |Gemma|w/o CD|18.16|
> ||lm-format|11.75|
> ||Outlines|13.56|
> ||Xgrammar|17.72|
> ||Formatron|18.02|
> |Llama3|w/o CD|31.75|
> ||lm-format|20.51|
> ||Outlines|23.71|
> ||Xgrammar|30.42|
> ||Formatron|30.42|
> |Qwen|w/o CD|35.40|
> ||lm-format|16.23|
> ||Outlines| 25.98|
> ||Xgrammar|34.34|
> ||Formatron|34.46|
>
> All constrained decoding methods require additional computation, of which our **Formatron remains the fastest**, almost approaching the speed of w/o CD.

---

### Official Review · Reviewer_YX9o · 2025-03-14

**Overall Recommendation:** 3

**Summary:**

This paper is about a novel method for grammar constrained decoding. Grammar constrained decoding poses many challenges to auto-regressive language model decoding, and as such a primary concern is to make it more efficient. This paper presents Formatron, an algorithm which keeps track of which states are still relevant, with the goal to make grammar-constrained decoding more efficient. It uses dynamic pruning based on the Earley algorithm. In particular, authors introduce “ZapFormat,” a method for tracking dependencies and removing unnecessary items. Experiments are provided, showing that the throughput is greatly increased using Formatron over other baseline methods.

**Claims And Evidence:**

The claims made in the submission are indeed supported by clear and convincing evidence. However, it would be informative to also report on the memory consumed by the various methods and their baselines (or alternatively, to give some analysis of how the memory should scale). This is because one of the primary claims is that this method reduces the memory overhead as well.

**Essential References Not Discussed:**

n/a

**Experimental Designs Or Analyses:**

See below.

**Methods And Evaluation Criteria:**

The benchmark datasets and the baseline methods do make sense for the problem at hand. In order to enhance clarity of Section 4, it would be useful to also include pseudocode. There are many components and it would be nice to present how they all interact with each other.

**Other Comments Or Suggestions:**

See above.

**Other Strengths And Weaknesses:**

Strengths:
* This paper adapt a well-known algorithm to language models. There are two issues with the Earley algorithm when it comes to decoding: one is that the required amount of sets to store scales with the length of the input, and the other is that the parser does not get rid of previous sets. This paper analyzes the challenges imposed by the Earley algorithm in LM decoding.
* This paper introduces an online method which is well-suited for inference both because it is more efficient (by dynamically pruning items) and requires less memory overhead.
* Experiments show that throughput is much faster with the proposed algorithm than three other methods on four tasks. Ablations are included as well.

Weaknesses:
* While it makes sense to report on the throughput of tasks on Task 1, (as this is one of the main goals of efficient GCD) in order to show that the method is also performing well qualitatively, it would nice to also report on some metric of the output such as accuracy or perplexity.
* The Formatron algorithm’s throughput degrades at 3 runs as seen in Table 2. Why is this the case? Is there anything different occurring in these different runs?
* Due to the claims of reduction in memory consumption, it would be insightful to share empirical details or other analysis regarding this claim.

**Questions For Authors:**

See above.

**Relation To Broader Scientific Literature:**

Methods like verifiers and grammars are in general becoming more popular for language models. It appears as though some sort of supervision may be useful. Hence, methods to improve efficiency for such methods should allow the literature to continue to explore these directions with a more scalable approach.

**Theoretical Claims:**

N/a

---

> ### Author Rebuttal · Authors · 2025-03-31
>
> ## Q1: In order to enhance clarity of Section 4, it would be useful to also include pseudocode.
>
> We appreciate the reviewer's valuable suggestion regarding Section 4. We agree that including pseudocode would enhance the clarity of this section. We will incorporate pseudocode in the revised manuscript.
>
> ## Q2: It would be nice to also report on some metric of the output such as accuracy or perplexity.
>
> Thank you for your insightful suggestion. We have evaluated output quality using accuracy metrics. Our experiments show that the proposed method achieves competitive performance compared to previous methods. We will incorporate this additional analysis in the revised manuscript.
>
> | Model  | Methods   | json_schema | json_grammar |
> |--------|-----------|-------------|--------------|
> | gemma  | baseline  | 0.73        | -            |
> |        | lm-format | 0.74        | 0.71         |
> |        | Outlines  | 0.80        | -            |
> |        | Xgrammar  | 0.76        | 0.71         |
> |        | Formatron | 0.73        | 0.74         |
> | Llama3 | baseline  | 0.47        | -            |
> |        | lm-format | 0.60        | 0.40         |
> |        | Outlines  | 0.73        | -            |
> |        | Xgrammar  | 0.69        | 0.47         |
> |        | Formatron | 0.67        | 0.48         |
> | Mistral| baseline  | 0.09        | -            |
> |        | lm-format | 0.53        | 0.10         |
> |        | Outlines  | 0.44        | -            |
> |        | Xgrammar  | 0.53        | 0.09         |
> |        | Formatron | 0.52        | 0.11         |
>
> ## Q3: The Formatron algorithm's throughput degrades at 3 runs as seen in Table 2. Why is this the case? Is there anything different occurring in these different runs?
>
> Thank you for your insightful question. The throughput variation in our initial 3 runs was likely due to resource contention on the shared server where we conducted experiments. After re-running tests on an idle server, we observed consistent improvements (see updated table). These new results confirm Formatron's throughput are stable. We will ensure all results in our revised paper are free from resource contention.
>
> | Model   | Methods   | 1 run     | 3 run     | 5 run     | 10 run    |
> |---------|-----------|-----------|-----------|-----------|-----------|
> | gemma   | Formatron | 7453.87   | 10180.72  | 10900.42  | 11929.77  |
> | Llama3  | Formatron | 7616.25   | 10271.73  | 11018.91  | 11945.94  |
> | Mistral | Formatron | 12828.53  | 13522.77  | 13841.85  | 14424.69  |
> | qwen    | Formatron | 8449.92   | 10537.26  | 11397.31  | 12202.48  |
>
> ## Q4: Due to the claims of reduction in memory consumption, it would be insightful to share empirical details or other analysis regarding this claim.
>
> Thank you for your important question regarding memory usage. We conducted additional experiments on max memory usage(unit: MB) of the process during constrained decoding. We note that pruning does help reducing memory usage. We will include these results in the revised manuscript.
> | Model   | Method       | Max Memory Usage (MB) |
> |---------|--------------|------------------------|
> | Llama3 | Formatron    | 1635.92               |
> | | w/o pruning  | 1655.48               |
> | Mistral    | Formatron    | 1519.09               |
> | | w/o pruning  | 1530.77               |

---

### Official Review · Reviewer_KTEP · 2025-03-20

**Overall Recommendation:** 3

**Summary:**

This paper proposes using the Earley parsing algorithm to speed up constrained decoding (e.g., for requiring output to be valid json). While existing methods for constrained decoding require looping over all tokens in the model vocabulary to generate the "mask" which determines which tokens are vs are not allowed to be generated at a given position during decoding, the Early algorithm avoids this large computational expense using a state-tracking algorithm.  With this approach, it is able to attain up to 2x speedup relative to state-of-the-art constrained decoding methods like XGrammar.

**Claims And Evidence:**

I believe so.

**Essential References Not Discussed:**

Not sure.

**Experimental Designs Or Analyses:**

I saw no immediate issues with the experimental soundness.

**Methods And Evaluation Criteria:**

I believe so.

**Other Comments Or Suggestions:**

- It would be helpful to provide a bit more of a primer for the key ideas/notation used in the paper.

**Other Strengths And Weaknesses:**

Strengths:
- The speedup results look quite good relative to the baselines.
- Constrained decoding is an important problem.

Weaknesses:
- I think the background necessary for understanding this paper, as well as the core method, could be better explained, I had a hard time understanding it.
- I'm not sure how much algorithmic novelty there is here. Is this just an efficient implementation of an existing algorithm?

I currently am scoring this paper with "weak accept" given that the results seem impressive, but I have low confidence in this review since I am not familiar with the literature or SOTA methods for tackling this problem.

**Questions For Authors:**

Does this method have any important limitations? Does it work for batched inference as well?

**Relation To Broader Scientific Literature:**

I am not familiar with the literature on this method.

**Theoretical Claims:**

N/A

---

> ### Author Rebuttal · Authors · 2025-03-31
>
> ## Q1: I think the background necessary for understanding this paper, as well as the core method, could be better explained, I had a hard time understanding it.
>
> Thank you for this important feedback about the paper's accessibility. We acknowledge that the background and core methodology of our paper could be more clearly presented. Due to the 8-page limit, we had to balance background explanation with results. In our revision, we will carefully enrich the background section and methodology description, taking full advantage of the additional space available in the camera-ready version. Additionally, we will invite colleagues from diverse technical backgrounds to review our revised manuscript to ensure our explanations are accessible and clear.
>
> ## Q2: I'm not sure how much algorithmic novelty there is here. Is this just an efficient implementation of an existing algorithm?
>
> Thank you for this important question regarding the novelty of our approach.
>
> Our work builds upon the Earley algorithm, which is fundamentally a theoretical framework. Our contribution lies in the innovative adaptations required to transform this theoretical construct originally designed for languages parsing into a practical solution for language model applications, which required developing both theoretical observations and theory-backed algorithm modifications rather than simply optimizing an existing implementation through engineering.
>
> We have made several adaptations to bridge theory and practice:
>
> 1. We noted that constrained decoding only requires format recognition rather than obtaining complete parsing trees, implying the potential to prune states in a modified Earley algorithm.
> 2. We formalized the idea of high-level regular grammars, allowing us to show what kinds of substructures will lead to repetitive states in the context of format recognition and hence can be effectively pruned.
> 3. Based on these theoretical constructs, we developed a domain-specific pruning strategy to manage Earley states efficiently in the context of language model generation.
>
> ## Q3: It would be helpful to provide a bit more of a primer for the key ideas/notation used in the paper.
>
> We appreciate the reviewer's suggestion on how to improve readability. We will enhance the paper by:
>
> 1. Adding a comprehensive notation table for reference. The following table is a partial example.
>
> | Symbol | Description |
> |--------|-------------|
> | A, B, X, Y | Non-terminal symbols in context-free grammar |
> | a, c | Terminal symbols in context-free grammar |
> | α, β, γ | Sequences composed of terminal and non-terminal symbols |
> | ε | Empty string |
> | S[i], ..., S[n] | Sequence of Earley state sets, where S[i] contains items at position i |
> | (X → α • β, j) | Traditional Earley item notation, where • indicates current parsing position and j indicates starting position |
> | (A → α • β, i, j) | Extended Earley item notation, where span [i, j] captures β's coverage range |
>
> 2. Providing a more accessible introduction to key concepts before diving into technical details
> 3. Including intuitive examples for complex ideas
>
> These improvements will make the paper more accessible.
>
> ## Q4: Does this method have any important limitations?
>
> We thank the reviewer for this important question on our method's applicability.
>
> Our method has no fundamental algorithmic limitations. However, the primary challenge lies in the nature of constrained decoding approaches generally - can only be applied during inference time and cannot be directly integrated into the model training process. Integrating constraint-decoding directly into supervised and RL training represents a promising direction for future research.
>
> ## Q5: Does it work for batched inference as well?
>
> We thank the reviewer for this important question about our method's efficiency in large-scale inference settings.
>
> Yes, Formatron fully supports batched inference. Our dynamic pruning and state caching mechanisms operate efficiently across multiple concurrent requests:
>
> 1. Our state caching system maintains separate pruned state sets for each sequence in the batch, allowing independent parsing paths from different grammars while sharing the same underlying algorithmic optimizations.
> 2. When processing multiple inputs simultaneously, Formatron's memory efficiency benefits become even more significant, as the dynamic pruning reduces the aggregate memory footprint across all batch elements.
> 3. The context-independent token mask cache is particularly effective in batched scenarios, as the precomputed masks can be efficiently applied across multiple sequences that share the same grammar constraints.

---

### Official Review · Reviewer_xPX3 · 2025-03-21

**Overall Recommendation:** 4

**Summary:**

This paper proposes ZapFormat, a dynamic pruning strategy that extends the Early algorithm for CFG parsing by eliminating invalid or redundant states. ZapFormat can improve inference speed of LLMs in constrained decoding.

**Claims And Evidence:**

The claims are clear and the evidence is convincing.

**Essential References Not Discussed:**

No.

**Experimental Designs Or Analyses:**

Yes. The experiment designs are sound.

**Methods And Evaluation Criteria:**

Yes. They make sense.

I do have one question regarding throughput results. I wonder if they are just the throughput of parsing & masking or if they include both parsing and generation time. Because 5000+ token per second seems way too fast even for non-constrained decoding.

If they only include parsing and masking, I think it would be better if the author could also report overall throughput, so that the improvements can be put into context.

**Other Comments Or Suggestions:**

The style of subsubsection titles such as 4.2.1 seems mismatching with the other section titles.

The 4.3 4.4 and 4.5 sections seem irrelevant to ZapFormat. Maybe you should describe them elsewhere.

**Other Strengths And Weaknesses:**

N/A

**Questions For Authors:**

How much do you think the speed of constrained decoding matters for reasoning models which have a really long output (thought) before calling some function?

**Relation To Broader Scientific Literature:**

This paper is a continuation of the efforts made to speed up constrained decoding. The methods used in it are a combination of existing ideas and the authors' own proposal. The results look really significant.

**Theoretical Claims:**

N/A

---

> ### Author Rebuttal · Authors · 2025-03-31
>
> ## Q: I wonder if they are just the throughput of parsing & masking
>
> Thanks for raising this important point on result presentation. We clarify that the throughput results in our paper indeed only reflect the parsing and mask generation stages, not the entire pipeline. We isolated these components to provide precise performance analysis of our key technical innovations.
>
> You're right that entire pipeline performance measurement would be valuable. we conducted additional experiments on throughput for the entire pipeline, including the throughput without contrained decoding (w/o CD).
>
> |Model|Methods|Json/s|
> |-|-|-|
> |Gemma|w/o CD|18.16|
> ||lm-format|11.75|
> ||Outlines|13.56|
> ||Xgrammar|17.72|
> ||Formatron|18.02|
> |Llama3|w/o CD|31.75|
> ||lm-format|20.51|
> ||Outlines|23.71|
> ||Xgrammar|30.42|
> ||Formatron|30.42|
> |Qwen|w/o CD|35.40|
> ||lm-format|16.23|
> ||Outlines| 25.98|
> ||Xgrammar|34.34|
> ||Formatron|34.46|
>
> Since the entire pipeline result includes LLM calls and generation, the throughput will be much lower. As can be seen from the new experimental results, our **Formatron is still the most efficient** among all constrained decoding methods, almost close to of w/o CD.
>
> ## Q: The style of subsubsection titles such as 4.2.1 seems mismatching with the other section titles.
>
> Thank you for highlighting this formatting inconsistency. We appreciate your meticulous review of the document structure. We will ensure all section and subsection titles follow a consistent style throughout the paper in our revision.
>
> ## Q: The 4.3 4.4 and 4.5 sections seem irrelevant to ZapFormat. Maybe you should describe them elsewhere.
>
> Thank you for your valuable feedback on paper organization. We agree that these sections may not directly contribute to the ZapFormat. We will restructure the paper to place these sections in a more appropriate location that better serves the overall narrative flow of our work.
>
> ## Q: How much do you think the speed of constrained decoding matters for reasoning models which have a really long output (thought) before calling some function?
>
> Thank you for this excellent question regarding constrained decoding for reasoning models.
>
> Our approach is flexible: for reasoning models with extensive thought processes, we can selectively apply constraints only to the function call portions, thereby maintaining efficiency comparable to non-reasoning models since the long reasoning output does not matter in this case. Alternatively, constrained decoding ensures that for ensuring thinking tags match, particularly for smaller models (7B parameters or fewer) or when processing out-of-distribution inputs. This helps maintain the paired tag structure required for proper reasoning format. For this case, the speed of constrained decoding is even more important since the number of constrained tokens could be enormous, much larger than non-reasoning models. This specific area is relatively unexplored though.
>
> Hence, your question identifies an important research direction worthy of further exploration, especially when reasoning processes require specific formatting constraints and maintaining generation efficiency is necessary.

---

### Decision · Program_Chairs · 2025-05-01

**Decision:**

Accept (poster)

**Comment:**

This paper focus on improving LLM decoding speed in a grammar constrained environment. The paper leverages Earley parsing and proposes improvements on the Earley parsing algorithm to achieve efficient next possible token calculations. Experiment results show impressive improvement on throughput parsing and masking.

Reviewers generally agree that this paper provide a solid improvement on accelerating grammar constrained decoding. Although some concern about the experiment setting has been raised, the author has addressed them accordingly.